# Combined PIVKA II and Vimentin-Guided EMT Tracking in Pancreatic Adenocarcinoma Combined Biomarker-Guided EMT Tracking in PDAC

**DOI:** 10.3390/cancers16132362

**Published:** 2024-06-27

**Authors:** Antonella Farina, Valentina Viggiani, Francesca Cortese, Marta Moretti, Sara Tartaglione, Antonio Angeloni, Emanuela Anastasi

**Affiliations:** Department of Experimental Medicine, “La Sapienza” University of Rome, V. Le Regina Elena 324, 00161 Rome, Italy; antoeffe22@gmail.com (A.F.); valentina.viggiani@uniroma1.it (V.V.); cortese.1186450@studenti.uniroma1.it (F.C.); marta.moretti@uniroma1.it (M.M.); sara.tartaglione@uniroma1.it (S.T.); antonio.angeloni@uniroma1.it (A.A.)

**Keywords:** PDAC, EMT, biomarkers, PIVKA II, Vimentin

## Abstract

**Simple Summary:**

Protein Induced by Vitamin K Absence (PIVKA II), also known as Des-γ-Carboxy prothrombin (DCP), is an established tumor marker for hepatocellular carcinoma (HCC) and a good diagnostic performance has also been demonstrated for Pancreatic Ductal Adenocarcinoma (PDAC). Circulating PIVKA II has been recently associated in vitro with epithelial-to-mesenchymal transition (EMT) activation in pancreatic origin cells PANC-1. The aim of this manuscript is to investigate whether this association is also detectable in vivo. The matter of this research lies in the fact that the diagnostic performance of PIVKA II in PDAC, along with its association with the EMT program, suggests that this molecule could be considered an indirect indicator of EMT and, therefore, a marker of the acquisition of an aggressive phenotype.

**Abstract:**

“Background/Aim”: the current inability to diagnose Pancreatic Cancer Adenocarcinoma (PDAC) at an early stage strongly influences therapeutic strategies. Protein Induced by Vitamin K Absence (PIVKA II) showed an accurate diagnostic performance for PDAC. Since circulating PIVKA II has been recently associated with pancreatic origin cells with Vimentin, an epithelial-to-mesenchymal transition (EMT) early activation marker, the aim of this study was to investigate in vivo the combination between the two proteins. “Materials and Methods”: we assayed the presence of PIVKA II and Vimentin proteins by using different diagnostic methods. A total of 20 PDAC patients and 10 healthy donors were tested by Western Blot analysis; 74 PDAC patient and 46 healthy donors were assayed by ECLIA and Elisa. “Results”: Western Blot analysis showed the concomitant expression of PIVKA II and Vimentin in PDAC patient sera. Immunometric assay performed on a larger cohort of patients demonstrated that 72% of PIVKA II-positive PDAC patients were Vimentin-positive. Additionally, in a group of PDAC patients with PIVKA II levels ≥2070 ng/mL, the percentage of Vimentin-positive subjects reached 84%. “Conclusion“: the association between PIVKA II protein and the EMT suggests that this molecule could be considered a marker of the acquisition of an aggressive phenotype.

## 1. Introduction

Pancreatic cancer (PC) is a malignant neoplasm that ranks as the seventh leading cause of cancer death worldwide. The most common type (85% of cases) of PC is the Pancreatic Ductal Adenocarcinoma (PDAC) that originates from the exocrine portion of the pancreatic gland [1]. According to GLOBOCAN ranking, only 5–7% of patients survive 5 years after diagnosis, and surgical resection is the only effective therapeutic treatment, with a 5-year survival rate [2]. Unfortunately, due to the absence of specific symptoms, 80% of patients are diagnosed at an advanced stage of the disease when the tumor can no longer be surgically treated [3]. Moreover, the risk of recurrence in surgically resected patients is extremely high (66 to 80%) [4,5]. Taking into account these considerations, early detection of the disease is crucial for improving prognosis [6]. In this perspective, integrating alternative diagnostic approaches into clinical practice, healthcare professionals can expedite the diagnostic process, initiate appropriate treatments sooner, and ultimately improve patient outcomes.

Serum biomarkers, readily available, seem to respond to this need. However, due to the poor sensitivity and specificity of suggested biomarkers such as Ca19.9, CEA, and CA242, research is constantly striving to identify new ones with better diagnostic performance in PDAC [7].

Among the emerging circulating biomarkers, Protein Induced by Vitamin K Absence (PIVKA II), also known as Des-γ-Carboxy prothrombin (DCP), an established tumor marker for hepatocellular carcinoma (HCC), is the most promising [8,9,10]. PIVKA II diagnostic potential in detecting PDAC has been recently highlighted by several studies, underscoring the relevance of its performance, especially compared to suggested serum biomarker guidelines (CA19-9, CEA, and CA242) which have been used in the management of this neoplasia [11,12,13,14,15]. It is well assessed that PIVKA II plays a crucial role in the development of various gastric tumors. Several studies have reported its involvement in hepatocellular carcinoma (HCC), where it promotes tumor growth and invasiveness by stimulating cell proliferation, remodeling of the extracellular matrix, and tumor angiogenesis [16,17]. To date, the mechanisms involving PIVKA II are not yet fully understood, although it has been reported that the cytoskeletal changes occurring during epithelial–mesenchymal transition (EMT) play a crucial role in PIVKA-II production through the impairment of vitamin K uptake [18].

EMT is a temporary and reversible cellular trans-differentiation program with a pivotal role in tumorigenesis; it is controlled by complex interactions among multiple signaling pathways such as TGF-beta, Wnt, and Notch, which converge in the activation of a network of specific Transcription Factors (EMT-TFs) such as Slug, Snail, Twist, and Zeb [19,20,21,22]. These factors, in turn, act pleiotropically on a series of genes that control various aspects of cellular physiology, repressing progressively epithelial characteristics and activating those typically mesenchymal [23,24]. The acquisition of mesenchymal phenotypes enables neoplastic cells to complete many steps along the evolutionary sequence of tumor progression [25]. Among biomarkers tracking the EMT signature, Vimentin is widely recognized as a canonical marker since it is upregulated during the neoplastic transformation of epithelial cells and its expression correlates with aggressiveness and an unfavorable prognosis [24,26,27]. This observation has also been supported by recent findings reporting that Vimentin expression is a potential independent adverse prognostic molecular marker and should be included in histopathological reports [28].

Although it has been reported that the presence of Vimentin on the surface of PDAC cells could be considered a potential marker, there are no data regarding circulating levels of Vimentin in PDAC [28,29]. A translational study demonstrated that PIVKA II is released by PANC-1 pancreatic cells following treatment with increasing doses of glucose. Interestingly, this release occurs concurrently with the activation of EMT [30], thus suggesting that in this model, the release of this biomarker in PDAC could be associated with a malignant phenotype.

In light of these observations and with the aim of translating these findings from bench to bedside, we sought to investigate whether there could be an association between circulating levels of PIVKA II and Vimentin in PDAC patients. For this purpose, we analyzed sera from a selected cohort of PDAC patients using Western Blot analysis and immunometry to detect the combined presence of PIVKA II and Vimentin.

## 2. Materials and Methods

### 2.1. Patients and Sera

This cross-sectional study included patients recruited at the Tumor Markers laboratory of Policlinico Umberto I, Rome, Italy, from September 2022 to October 2023.

We analyzed 120 serum samples subdivided as follows:74 from PDAC patients;46 healthy blood donors.

PDAC patients met the following eligibility criteria: the first occurrence of neoplastic pathology, no prior treatment with neoadjuvant therapy, absence of diabetes, no serious physical disabilities. Patients were excluded if their alcohol consumption was higher than a single serving a day, if they showed evidence of an active hepatopathy, if they took anti Vitamin K antagonists, or if they had any coagulopathy. At enrollment, medical history was collected for each subject, and peripheral blood samples were drawn and immediately sent to the laboratory of Tumor Markers of Policlinico Umberto I, Rome, Italy. PDAC was diagnosed by histology or by imaging methods (multiphasic computed tomography or dynamic contrast-enhanced magnetic resonance); when the presence of the aforementioned neoplasms was confirmed, each serum sample was analyzed for PIVKA-II [11,12]. The PDAC patients and healthy control cohorts had similar demographics in terms of ratio of men to women, with the majority of all subjects being Caucasian. Healthy donors were recruited from the biobank “Centro Trasfusionale” of the “Azienda Ospedaliera Policlinico Umberto I”, La Sapienza University of Rome.

All study participants were over 18 years of age and signed a written informed consent form for the investigation voluntarily before enrollment. The study protocol was approved by Policlinico Umberto I Review Board and was performed in accordance with the Declaration of Helsinki.

### 2.2. Blood Sampling

Blood collection was performed following a standard protocol [11]. After signing the informed consent form for participation in the study and acquisition of personal pathological anamnesis, all patients underwent peripheral venous puncture; blood samples were collected in a yellow top Vacutainer, and after clotting for 60–90 min, were centrifuged for 10 min at 1300× *g*. Following the collection of the serum samples, an amount of 500 µL of each serum fraction was aliquoted in Eppendorf tubes and stored at −80 °C until analysis.

### 2.3. Western Blot Analysis

Selected human sera were diluted 1:360 in 1X PBS and 4X SDS Sample loading buffer. A total of 20 µL of diluted sera was loaded on 10% TGX FastCast according to the manufacturer’s instruction. Separated proteins were then transferred to nitrocellulose membranes for 45 min in Tris-glycine buffer and immunoblotted, as described elsewhere [31]. Membranes were first stained with Red Ponceau then probed with anti PIVKA-II (1:1000), anti-Vimentin (1:200), polyclonal anti-mouse IgG-HRP (1:10,000). Detection was performed using Western Bright (Advansta, Menio Park, CA, USA) following manufacturer instruction [30].

### 2.4. Immunometry

Serum PIVKA II levels were assessed in ng/mL with the Elecsys PIVKA-II kit on Roche^®^ COBAS e411 (Basel, Switzerland), a fully automated analyzer based on a one-step sandwich assay with electrochemiluminescence (ECLIA) technology, as described elsewhere [32]. Elecsys PIVKA-II immunoassay is characterized by a detection range between 3.5 and 12,000 ng/mL, while the LoD and LoQ were, respectively, ≤3.5 ng/mL and ≤4.5 ng/mL [33]. The instructions of the manufacturer were followed to perform the assays. All tests were conducted in duplicate. Serum Vimentin levels were measured using a two-step sandwich manual ELISA assay, employing the Human Vimentin (VIM) ELISA kit following manufacturer instruction. All tests were conducted in duplicate.

### 2.5. Statistical Analysis

Descriptive statistics were used to calculate mean or number (percentage) of the study population. The diagnostic accuracy of PIVKA II was evaluated by receiver operating characteristic (ROC) curve analysis. PIVKA II performance was reported as area under the curve (AUC). For AUC, we estimated the 95% confidence interval (95% CI). The relationship between PIVKA II positivity and clinico-pathological parameters (healthy vs. diseased) was analyzed using the non-parametric Mann–Whitney test. The statistical significance of the difference in Vimentin positivity rates across various PIVKA-II concentration intervals in PIVKA-II-positive patients was determined using the Chi-square test. We considered statistically significant a two tailed *p*-value < 0.05. All statistical analyses were performed using StatsDirect 3.0.187 statistical software (StatsDirect software, Cheshire, UK).

## 3. Results

### 3.1. Western Blot Analysis Exhibits the Presence of Circulating Levels of Vimentin and PIVKA II in PDAC Patient Sera

We sought to investigate, by Western Blot analysis, the presence of circulating Vimentin and PIVKA II in the sera of PDAC patients. Therefore, we separated by SDS-PAGE 20 μL of human sera (1:320) from a selected cohort of PDAC patients. Figure 1 shows a representative Western Blot performed using the specific antibody directed against PIVKA II and subsequently against Vimentin. As negative controls (−), we use the sera of a healthy subject. Since it is not feasible to quantify serum proteins due to the absence of a normalizer, to ensure proper sample loading, we stained the membranes with Red Ponceau, and the most prominent band observed was albumin. We observed in all three PDAC samples a 67 kDa band, while a 57 kDa band corresponding to Vimentin was visible in two out of three of the PDAC sera. None of the healthy donors exhibited detectable levels of PIVKA II or Vimentin. Results observed from the study of human sera by WB (20 PDAC patients and 10 healthy donors) are summarized in Table 1.

### 3.2. Vimentin Threshold in Human Sera

A receiving operating characteristic (ROC) curve was constructed to provide the optimal cut-off for discriminating between healthy individuals and affected subjects; the ROC curve identified a cut-off of 487 ng/mL, obtained with an AUC of 74%, a sensitivity of 73%, and a specificity of 72% (95% CI) Figure 2a.

### 3.3. Comparative Analysis of Vimentin and PIVKA II in Human Sera

Based on the suggested cut-off value, we then analyzed circulating levels of Vimentin in a cohort of 74 PDAC patients, compared to 46 healthy donors. As shown in Figure 2b, we observed that 72% of PDAC patients (53 out of 74) were Vimentin-positive. We also evaluated/tested PIVKA II levels in PDAC patients compared to healthy donors. Since PIVKA II is an assessed PDAC biomarker, we considered the suggested threshold >30 ng/mL. In this case, we observed that 90.5% of the PDAC population exhibited increased levels of this circulating marker.

### 3.4. Vimentin-Positive Sera Correlate with High PIVKA II Levels in PDAC Patients

Then, we analyzed the association between the two proteins in PDAC patients considering a bimodal assignment for Vimentin (positive/negative). Therefore, we arbitrarily divided the PIVKA II-positive patients as follows:

Group I: PIVKA-II levels ranging from 31 to 2069 ng/mL;Group II: PIVKA-II levels ranging from 2070 to ≥12,000 ng/mL.

We observed that in Group I, 62% of patients were Vimentin-positive, whereas in Group II, the percentage of Vimentin-positive subjects reached 84% (*p*-value = 0.0578). These results are summarized in Figure 3.

## 4. Discussion

In the present study, we analyzed the presence of PIVKA II in PDAC patients using different diagnostic methods: the Western Blot analysis and immunometry. Western Blot analysis, often used to separate cellular lysate and rarely human sera, is considered a gold standard technique for protein characterization, but high costs, the requirement of highly specialized operators, and the limited number of samples that can be analyzed per run represent significant limitations when attempting to test a larger population. Thus, we shifted to more conventional immunoassay techniques. Since reference values and the decisional threshold for Vimentin in PDAC were not established, we calculated with the Receiving Operating Curve an optimal cut-off (487 ng/mL). Considering this threshold, we found altered levels of Vimentin in 72% of PDAC patients. The correlation between PIVKA II and Vimentin in PDAC patient sera was further defined by comparing Vimentin positivity with PIVKA II circulating levels in two different groups arbitrarily subdivided based on PIVKA II concentration. Although the difference between the two groups was not highly significant (*p*-value = 0.0578), we observed a trend towards significance between Vimentin positivity and high PIVKA-II levels (above 2070 ng/mL). It is plausible that increasing the sample size could strengthen this finding. We can speculate that if this incremental trend is confirmed, a non-random correlation between the two proteins could be derived, especially when PIVKA II levels are above the discriminatory threshold set at 2070 ng/mL. While the Chi-square model has been instrumental in analyzing the relationships between categorical variables in our study, we acknowledge its limitations, particularly its inability to control for confounding variables. This limitation means that the Chi-square model cannot account for external factors that might influence the observed associations, potentially affecting the validity of our findings. Despite these shortcomings, the Chi-square model was chosen for its simplicity and efficiency in initial exploratory analyses. It provides a straightforward method to identify potential associations that can be further investigated with more complex models. By utilizing the Chi-square test, we were able to quickly identify significant relationships that warranted deeper analysis. Further validation studies with larger sample sizes and diverse populations could help mitigate these methodological biases and enhance the robustness of our findings.

Data obtained so far, together with previous in vitro evidence, highlight the close relationship between the production and release of PIVKA II and the activation of EMT. These events are undoubtedly interrelated and can thus be considered potential driving forces in the evolutionary path that a neoplastic cell undergoes towards malignancy.

PC is known for its early and aggressive tendency of local invasiveness as well as its high metastatic potential. These characteristics, combined with the late onset of symptoms, render it a malignancy with generally unfavorable prognosis, high mortality, and with a 5-year survival rate of 9% [34].

Serum biomarkers, being readily available and obtainable at low cost with high patient compliance, represent a powerful tool both for detecting early stages of the disease and for managing follow-up and surveillance. However, the molecules suggested to date as biomarkers for PDAC still demonstrate their clinical diagnostic inefficacy, in terms of reduced sensitivity and specificity [14].

Growing attention has been focused in clinical trials on identifying a cost-effective biomarker with better sensitivity and specificity to improve PDAC early diagnosis, and among these, the contribution of the protein PIVKA II stands out prominently. Over the last decade, several studies have revealed the presence of altered PIVKA II levels in several gastrointestinal neoplasia, and this is especially true for HCC, since increased levels of this protein are often correlated with the tumor dimension, with the metastatic potential, and with tumor relapse [35,36]. Interestingly, in PDAC, PIVKA II serum levels have been found to be significantly elevated with respect to benign pancreatic diseases, thus suggesting its high diagnostic potential. This finding have also been supported by the evidence that PIVKA II is decreased after surgery in PDAC patients [12,37,38]. Further in vitro studies have also highlighted the possible role of PIVKA II in carcinogenesis, demonstrating that the release occurs in a glucose dose-dependent manner jointly with the activation of EMT [30]. Notably, a similar association between PIVKA II and EMT was also observed in 2011 by Suzuki et al., who reported that in the context of HCC, HepG2 cells produce PIVKA II when vitamin K uptake is impaired by the cytoskeletal rearrangement that occurs during the phenotypic changes associated with EMT [39].

The activation of the EMT program is considered one of the hallmarks of the tumor and a leading cause of death in patients with solid tumors [40]; thus, predicting tumor metastasis could help with the implementation of personalized therapy in the clinical treatment of tumors, leading to better outcomes for cancer patients. The acquisition of an EMT-driven mesenchymal phenotype enables the invasion of surrounding tissues, formation of distant metastases, metabolic reprogramming, resistance to chemotherapeutics, and suppression of the immune system [41]. In PDAC, the activation of this program represents one of the first events that occurs at a temporal level, giving to the tumor its characteristic aggressiveness [38,42,43]; thus, identifying metastatic biomarkers helps to detect initial tumor metastasis or recurrence, thereby improving the treatment and management strategy for cancer patients.

To better address the clinical aspects of our study, it is crucial to highlight that the use of biomarkers is highly dependent on their sensitivity and specificity. Previous studies have demonstrated that in PDAC patients, PIVKA-II showed an optimal AUC with a sensitivity of 92% and specificity of 80%, outperforming other suggested standards for pancreatic cancer [37]. Combining PIVKA II with CA19-9 improves diagnostic accuracy significantly (AUC of 0.945, sensitivity of 87.7%, specificity of 94.4%) [37]. While our data indicate that PIVKA II is less likely to elevate above the cut-off in benign pancreatic diseases compared to CA19-9, CEA, and CA242 [37], the potential for false positives still exists. This is particularly relevant in clinical practice where diagnostic accuracy is critical to avoid unnecessary treatments or delays in appropriate care. The clinical utility of Vimentin relies on its suggested value as a histological biomarker in PDAC due to its surface expression on circulating tumor cells (CTCs). Studies [44,45] have reported that combining Vimentin, CTCs, and CA19-9 provides favorable diagnostic potency. Aware of the limitations, our study highlights how the combination of PIVKA II and Vimentin could facilitate the early diagnosis of PDAC, which is crucial for timely therapeutic intervention.

## 5. Conclusions

The promising diagnostic performance of PIVKA II in PDAC, along with its association with the EMT program, suggests that this molecule could be considered an indirect indicator of EMT and, therefore, a marker of the acquisition of an aggressive phenotype. Further studies are indeed necessary to delve deeper into this topic.

## Figures and Tables

**Figure 1 cancers-16-02362-f001:**
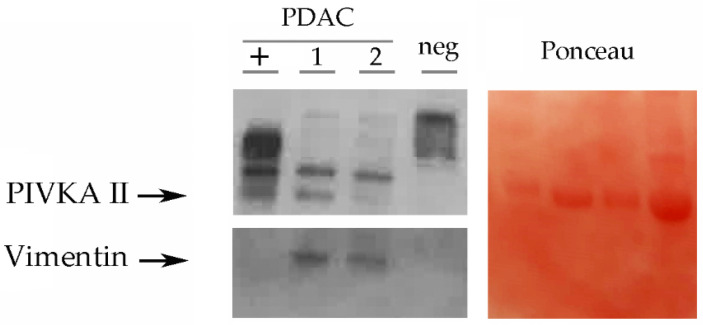
PIVKA II and Vimentin expression in human sera. Representative Western Blot analysis performed on human sera. Sera were treated as described in Materials and Methods. On the **left** panel, PDAC sera (1–2–3) showed PIVKA II (67 kDa) and Vimentin (57 kDa). The presence of upper migrating bands is a not specific signal. Healthy donor serum was loaded as a negative control (neg). When loading control, we show a Ponceau staining of the blot (**right** panel), and the prominent band of this staining is albumin. One representative experiment out of three is shown.

**Figure 2 cancers-16-02362-f002:**
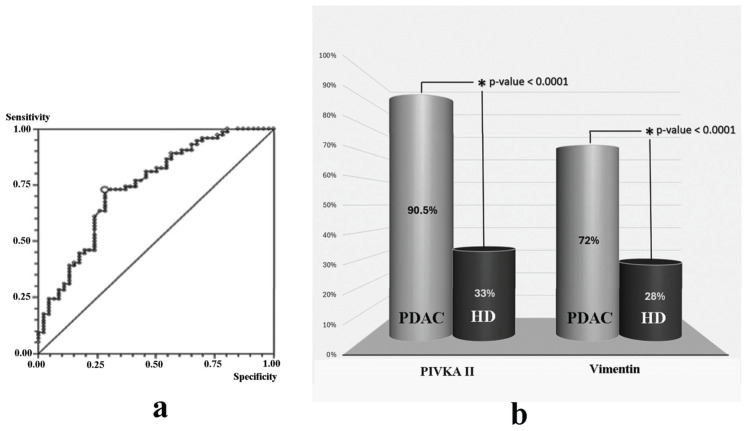
Vimentin and PIVKA II levels in PDAC and HD. (**a**) Receiving operating characteristic (ROC) curve on Vimentin. Suggested threshold is 487 ng/mL. (**b**) Vimentin percentage of positivity in healthy donors compared to PDAC patients. HD = healthy donor; PDAC = Pancreatic Adenocarcinoma sera.

**Figure 3 cancers-16-02362-f003:**
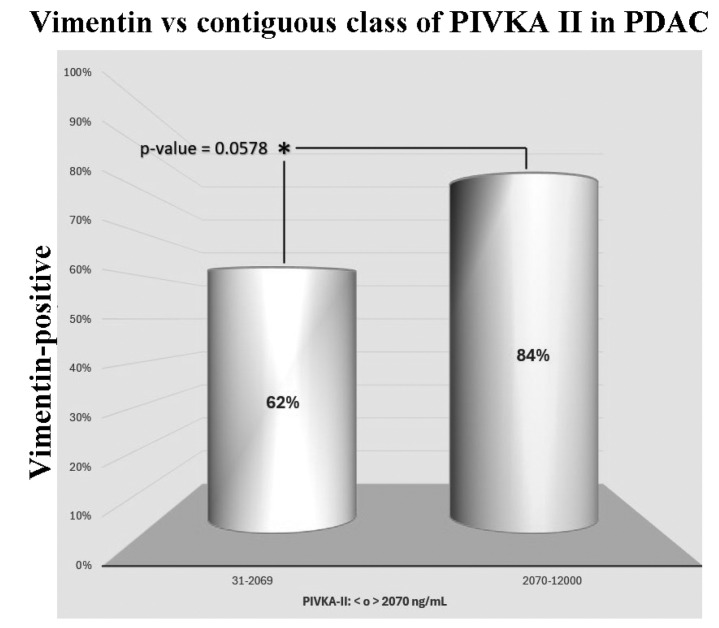
Distribution of Vimentin-positive PDAC patients in contiguous classes of PIVKA II.

**Table 1 cancers-16-02362-t001:** Screening of human sera by WB analysis.

Sera	N°	PIVKA II +	Vimentin +
HD	10	nd	nd
PDAC	20	95%	85%

HD = healthy donor; PDAC = Pancreatic Adenocarcinoma sera; nd = not detected.

## Data Availability

Data are contained within the article.

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
