# Peer review of "Combined PIVKA II and Vimentin-Guided EMT Tracking in Pancreatic Adenocarcinoma Combined Biomarker-Guided EMT Tracking in PDAC"

_cancers, 2024, doi:10.3390/cancers16132362_

Round 1

Reviewer 1 Report

Comments and Suggestions for Authors

Methodological Biases exist

Title must be changed

Many References are missing

Discussion contains repetitions and irrelevant data

(The Authors must see my remarks)

Author Response

Thank you for your comments, they were useful to clarify and improve the quality of our research. According with your suggestions, we've made the requested modifications and prepared the manuscript for the resubmission. 

Reviewer 2 Report

Comments and Suggestions for Authors

The study by Farina et al. aims to demonstrate that Protein Induced by Vitamin K Absence (PIVKA II) can serve as a biomarker for tracking epithelial-to-mesenchymal transition (EMT) in Pancreatic Ductal Adenocarcinoma (PDAC). Using a combination of western blot analysis, ECLIA, and ELISA, the authors analyzed the serum levels of PIVKA II and Vimentin in a cohort of PDAC patients and healthy donors. However, the results are unconvincing due to conflicting outcomes, making the study unsuitable for publication in Cancers.

Comments:

1.      The research lacks a comparison with other established PDAC biomarkers, which is crucial for validating the efficacy of PIVKA II as a reliable biomarker.

2.      In Figure 1, the Ponceau blot does not provide clear or useful control information. The study lacks proper controls, such as well-known marker proteins and control protein immunoblotting, which are essential for validating the experimental results.

3.      The study tested 46 healthy donors and 74 patients. In Table 1, the authors reported zero positive results for both PIVKA II and Vimentin in 10 out of 46 healthy donors. However, in Figures 2b and 2c, the positive percentage increases to around 30%. This inconsistency raises questions about the reliability of PIVKA II as a biomarker and the potential rate of false positives.

4.      The study lacks comprehensive statistical analysis to support its findings, undermining the validity and robustness of the conclusions drawn.

In conclusion, the study by Farina et al. falls short in several critical areas, including comparative analysis, control validation, result consistency, and statistical rigor, making it unsuitable for publication in its current form.

Comments on the Quality of English Language

Overall, the English language in the draft is adequate, but some phrases are too conversational and not suitable for publication. Additionally, improvements are needed to enhance readability and ensure the text meets academic standards.

Author Response

(The authors gave the same response as above.)

Round 2

Reviewer 1 Report

Comments and Suggestions for Authors

Methodological Biases exist

Chi-square model is weak and unreliable (can not ''control'' possible confounders)

Author Response

Thank you for you comment, according to your observation we believe that the chi-square model is indeed limited in its ability to control for potential confounders, as it primarily assesses the association between categorical variables. Further validation studies with larger sample sizes and diverse populations could help mitigate these methodological biases and enhance the robustness of our conclusions.

Reviewer 2 Report

Comments and Suggestions for Authors

The authors' responses partially address the reviewers' concerns. Overall, the data now make more sense and better support the claims. However, the authors still need to discuss and highlight the sensitivity and selectivity of combining PIVKA II and Vimentin for diagnosis compared with existing, well-established clinical methods in the manuscript. Additionally, they should discuss the potential for false positives and the implications for clinical practice. This further discussion would enhance the manuscript's completeness and address the concerns regarding the reliability and clinical utility of the proposed biomarkers.

Author Response

Thank you for your kind suggestions. To this regard, we considered it necessary to integrate the manuscript discussion with the statements reported below.

To better address the clinical aspects of our study, it's crucial to highlight that the use of biomarkers is highly dependent on their sensitivity and specificity. Previous studies have demonstrated that in PDAC patients, PIVKA-II showed an optimal AUC with a sensitivity of 92% and specificity of 80%, outperforming other suggested standards for pancreatic cancer [39]. Combining PIVKA II with CA19-9 improves diagnostic accuracy significantly (AUC of 0.945, sensitivity of 87.7%, specificity of 94.4%)[39]. While our data indicate that PIVKA II is less likely to elevate above the cut-off in benign pancreatic diseases compared to CA19-9, CEA, and CA242 [39], the potential for false positives still exists. This is particularly relevant in clinical practice where diagnostic accuracy is critical to avoid unnecessary treatments or delays in appropriate care. Clinical utility of Vimentin relies on its suggested value as a histological biomarker in PDAC due to its surface expression on circulating tumor cells (CTCs). Studies [46, 47] have reported that combining Vimentin, CTCs, and CA19-9 provides favorable diagnostic potency. Aware of the limitations, our study highlights how the combination of PIVKA II and Vimentin could facilitate the early diagnosis of PDAC, which is crucial for timely therapeutic intervention.” (Lane 272-287).
